

# On classical and hybrid shadows of quantum states

**Saumya Shivam[1], Curt W. von Keyserlingk[2,3] and Shivaji L. Sondhi[4]**

**1** Department of Physics, Princeton University, Princeton, New Jersey 08544, USA
**2** School of Physics & Astronomy, University of Birmingham, Birmingham, B15 2TT, UK
**3** Department of Physics, King's College London, Strand WC2R 2LS, UK
**4** Rudolf Peierls Centre for Theoretical Physics, University of Oxford,
Oxford OX1 3PU, United Kingdom

## Abstract

Classical shadows are a computationally efficient approach to storing quantum states on a classical computer for the purposes of estimating expectation values of local observables, obtained by performing repeated random measurements. In this note we offer some comments on this approach. We note that the resources needed to form classical shadows with bounded *relative error* depend strongly on the target state. We then comment on the advantages and limitations of using classical shadows to simulate many-body dynamics. In addition, we introduce the notion of a hybrid shadow, constructed from measurements on a part of the system instead of the entirety, which provides a framework to gain more insight into the nature of shadow states as one reduces the size of the subsystem measured, and a potential alternative to compressing quantum states.



# 1 Introduction

An idealized quantum computing platform allows one to: i) repeatedly create a complex quantum state by applying a specified quantum circuit to a simple initial state, ii) destructively measure the state in the computational basis, possibly applying further unitary gates before doing so. Given these capabilities, it is natural to ask how to go about efficiently and accurately capturing properties of the state (e.g., entanglement and correlation functions), while requiring as few preparations/measurement iterations as possible. Can one perform a strategically choose a set of measurements, and use the outcomes to create an approximation to the quantum state which is of reasonable size, and which can be used to estimate various state properties such as expectation values of a range of operators? Any such procedure would also be of purely theoretical interest in that it would furnish a compressed description of quantum states that could perhaps be used in classical computations, much as matrix product states are used to efficiently compute the static and dynamic properties of $d = 1$ quantum systems with low entanglement.

A recent development along these lines goes under the moniker of shadow tomography. The term itself dates from the work of Aaronson [1] who investigated how many samples would be needed for the task of reconstructing a set of expectation values to a given accuracy. Huang *et. al.* [2] took inspiration from this work to define the notion of a classical shadow of a quantum state which is a set of approximations to the state constructed from snapshots in which multiple copies of the quantum state are prepared and measured (in a randomized basis). In the limit of an infinite number of snapshots, averages over the shadow equal quantum mechanical averages in the exact state, and in certain cases surprisingly few snapshots are required to accurately approximate a large set of observables, *which need not be specified in advance*. The body of results surrounding these ideas is the subject of a recent review [3]. We note that the simplest example of a classical shadow, which we will use below, was introduced earlier by Paini and Kalev [4] albeit without naming it as such.

In this note we offer some comments on classical shadows from the viewpoint of quantum many body theory and statistical mechanics. We are interested in the nature of the representation of the density matrix that lies at the heart of shadow technology, the utility of shadows for measuring quantities of particular physical interest, for reducing the effort involved in time evolving many body quantum systems on a classical computer and in a generalization to hybrid classical-quantum shadows that interpolates between the full quantum state and its classical shadow representation.

# 2 Basics

Huang et al. [2] gave a general construction of classical shadows based on ensembles of random unitaries. In their construction, a unitary $U$ (drawn randomly from the ensemble) is applied to the state $\rho$, yielding $U\rho U^{\dagger}$. Then one measures each qubit in the $Z$ basis and

constructs a snapshot of the state as the operator $U^\dagger |b\rangle \langle b| U$, where $|b\rangle$ is the measurement outcome. Averaging snapshots over the unitaries and the measurement outcomes defines a linear map $\mathcal{M}(\rho)$

$$\sum_b \int dU U^\dagger |b\rangle\langle b| U \langle b|U\rho U^\dagger|b\rangle = \mathcal{M}(\rho), \tag{1}$$

from density matrices to density matrices (since $\mathcal{M}(\rho)$ is a completely positive trace preserving map). Here $dU$ is some as yet unspecified measure on the space of many-body unitary matrices, and defines the ensemble.

If the ensemble of unitaries is chosen such that the map $\mathcal{M}$ is invertible, then the inverse map applied to Eq. (1) provides an exact expression

$$\rho = \sum_b \int dU \mathcal{M}^{-1}(U^\dagger |b\rangle\langle b| U) \langle b|U\rho U^\dagger|b\rangle, \tag{2}$$

for $\rho$ as a linear combination of the basis operators $\mathcal{M}^{-1}(U^\dagger |b\rangle \langle b| U)$ with coefficients that form a probability distribution and further are simply the Born rule probabilities to obtain outcome $|b\rangle$ in a single measurement following the application of unitary $U$ the state. Following standard classical statistical reasoning we can now sample from this distribution and for a finite number of samples $M$, obtain the estimate $\hat{\rho}_s = \frac{1}{M}\sum_{m=1}^{M} \hat{\rho}_m$ for the state. The set $\{\hat{\rho}_m\}$ is referred to as the *classical shadow* of the quantum state and each $\hat{\rho}_m$ is an *inverted snapshot*. It is important to note that the basis operators or inverted snapshots are not legitimate density matrices—while they are hermitian and have unit trace, they are *not* positive semi-definite.

When $U$ is a global Clifford or Haar random unitary, the inverse can be written as [2]

$$\mathcal{M}^{-1}(U^\dagger |b\rangle \langle b| U) = (2^L + 1)U^\dagger |b\rangle \langle b| U - \mathbf{I}_{2^L \times 2^L}, \tag{3}$$

with all but one eigenvalues being negative. The inverted snapshots can then be used to efficiently estimate the expectation values of operators with a bounded Hilbert-Schmidt norm, but are ill suited for typical few body observables that we consider[1] [2]. Further, given that global Clifford unitaries are less reliably implemented on near term quantum computers, we restrict our discussion to local unitaries for practical considerations, while generalizing our claims to global unitaries in an Appendix.[1]

The special case of $U$ being a tensor product of independent Haar random unitaries on each qubit was studied by Paini and Kalev [4] earlier in the equivalent language of picking random measurement axes. Here the inverse exists and takes the form

$$\mathcal{M}^{-1}(U^\dagger |b\rangle \langle b| U) = \bigotimes_{j=1,\ldots,L} \left(3U_j^\dagger |b_j\rangle \langle b_j| U_j - \mathbf{I}_{2\times 2}\right), \tag{4}$$

where we see that half of the eigenvalues are negative. In fact one obtains the same inverse map when independent single site random Clifford unitaries alone are used in place of random Haar unitaries [2].

Given an operator $O$ (assumed without loss of generality to be traceless), the classical shadow can be used to estimate the expectation value $o = \text{Tr}(\rho O)$. Each element of the shadow yields a random variable $\hat{o}_m \equiv \text{Tr}(\hat{\rho}_m O)$, which averages to $o$. The $\hat{o}_m$ are identically distributed and independent for different $m$; we refer to a single such random variable as $\hat{o}$. The $\hat{o}_m$ may be combined in various ways to estimate $o$. For example one could take the average $\sum_{m=1}^{M} \hat{o}_m / M$. Alternatively, one can take averages of subsets of $\{o_m\}_{m=1}^{M}$, and then take the median of those averages (median of means). This latter method turns out to

---

[1]See supplementary material.

preferable. For appropriately chosen size of subsets, the $M$ required to achieve an accuracy $\epsilon$ with probability $> 1 - \delta$ is given by

$$M \geq C \frac{\mathrm{var}(\hat{o})}{\epsilon^2} \log(2/\delta), \tag{5}$$

where $C$ is an unimportant numerical constant [2]. It is clear that the number of samples required is controlled by the variance of the shadow random variable $\hat{o}$.

## 3 Comments

We now offer some comments on the formalism described above. Note that for simplicity we restrict our discussions considering $O$ to be a Pauli string, and avoid the more general problem of simultaneously estimating the error for a number of different $O$ [2, 5], for which the qualitative nature of our arguments would not be different.

### 3.1 Random and non-random measurements

Let us specialize to the case where the operator $O$ is a $k$-body Pauli string. It is possible to calculate the variance in Eq. (5) exactly for the Clifford ensemble [2], and the result is

$$\mathrm{var}(\hat{o}) = 3^k - \rho(O)^2. \tag{6}$$

Therefore the shadow recipe above requires $M \geq \frac{C}{\epsilon^2}(3^k - \rho(O)^2) \times \log(2/\delta)$ samples in order to guarantee that the absolute error in our estimate of $\langle O \rangle$ is less than $\epsilon$. However suppose we happen to know in advance that $O$ is a $k$-body string of Pauli $Z$ operators alone. Can we do better? The answer is yes, of course. In this case just estimate $O$ in the usual way: always measure the state in the computational ($Z$) basis. For $M$ experimental runs, this generates a list of outcomes $\{\hat{o}'_m\}_{m=1}^M$, each of which takes values in $\{-1, 1\}$. Once again we may estimate $o$ by taking the average outcome across experimental runs (using median of means method). The $M$ required to achieve an accuracy $\epsilon$ with probability $> 1 - \delta$ is given by formula Eq. (5) with $\hat{o} \to \hat{o}'$. This makes a dramatic change, as it is easy to show that $\mathrm{var}(\hat{o}') = 1 - \rho(O)^2$ (cf. Eq. (6)). Therefore $M \geq \frac{C}{\epsilon^2}(1 - \rho(O)^2) \times \log(2/\delta)$ measurements suffice when we know in advance that $O$ is a string of Pauli $Z$ operators. Crucially, in this latter case the measurements required do not increase exponentially with $k$.

While we considered a $z$ string, clearly we will get the same bound for any specified Pauli string as we can simply measure along the appropriate axis on each site. But this strategy fails when we wish to estimate multiple and arbitrary Pauli strings on $k$ sites. Indeed we may estimate that in order to have any chance accurately estimating $\langle O \rangle$ for any such $k$-body Pauli string, we will need to have measured each of the $k$ sites along each of the 3 axes multiple times leading to $\mathcal{O}(3^k)$ measurements consistent with the factor of $3^k$ in the shadow bound. We note that the superiority of the traditional method for measuring the expectation value of fixed operators has been discussed in Ref. [5] under the terminology of "de-randomization" wherein you assume that random measurements are the place to begin.

In the case when one has access to two copies of the unknown state, denoted by $\rho \otimes \rho$, along with the ability to perform Bell measurements across the two copies, the absolute expectation value of $O$, $|o| = |\mathrm{Tr}[\rho O]|$ can be estimated using $M \geq \frac{C}{\epsilon^4} \times \log(2/\delta)$ measurements for any Pauli string $O$, owing to the fact that Bell states are eigenstates of any such $O \otimes O$ [6, 7]. While this approach has the advantages that $O$ need not be known beforehand, and that the number of measurements does not scale with the weight of the Pauli string, the disadvantages are that it is limited to $O$ being a Pauli string, and that the scaling with the accuracy $\epsilon$ is worse than

both classical shadows and de-randomized measurements ($1/\epsilon^2$). Moreover, to find the sign of the expectation value, additional measurements in the basis of $O$ may be needed (requiring prior knowledge of $O$) [6].

## 3.2 State dependence of absolute and relative error

Classical shadows provides an efficient approach to measure expectation value of an operator $O$ because of the independence of the bound on absolute error with the state $\rho$ and the system size $L$ (Eq.5). But often the relative error ($\sqrt{\text{var}(\text{Tr}[\hat{\rho}_s O])}/\text{Tr}[\rho O]$) can be important as well, and estimates using classical shadows have a strong state dependence. Consider $O$ to be a Pauli string of weight $k$, and a single site Clifford/Haar random unitary ensemble. Using our formula Eq. 6, the squared relative error is

$$\frac{\text{var}(\hat{o})}{\text{Tr}[\rho O]^2} = \left( \frac{3^k}{\text{Tr}[\rho O]^2} - 1 \right). \tag{7}$$

Therefore, the number of samples $M$ required to achieve the same accuracy $\epsilon$ and confidence interval $\delta$ is given by

$$M \geq \frac{1}{\epsilon^2} \left( \frac{3^k}{\text{Tr}[\rho O]^2} - 1 \right) \log(2/\delta). \tag{8}$$

The required $M$ can vary dramatically depending on the state $\rho$. For example, a Haar random state has $\text{Tr}[\rho O] = e^{-\mathcal{O}(L)}$, which would require an exponential in system size number of samples. On the other hand, if $\rho$ is a product state in $Z$ basis and $O$ contains only $Z_j$ acting on $k$ qubits individually, then $\text{Tr}[\rho O]^2 = 1$, so that $M$ is independent of system size. Another interesting example is when measuring the correlator $O = Z_i Z_j$ between two sites $i, j$ in an ordered/critical phase of a Hamiltonian where $\text{Tr}[\rho O] \sim \exp(|i-j|/\xi) / \sim |i-j|^{-d+2-\eta}$ and the number of samples $M$ correspondingly scales exponentially or polynomially as $|i-j|$ increases.

The state dependence of the relative error is numerically examined in Figure 1, where we compare the relative error in the expectation value of $O = Z_i$ for the classical shadow of a typical Haar random state. The relative error increases with $L$, while the variance remains approximately constant. We also consider the relative error in $Z$ acting only on the first qubit for the state $|\psi\rangle = \frac{1}{\sqrt{L}}(|0111\ldots\rangle + |1011\ldots\rangle + \ldots)$. Here the expectation value $\text{Tr}[\rho Z_1] = 1 - 2/L$ increases with $L$ sufficiently quickly that the relative error decreases with system size (in contrast with the Haar random case).

To summarize, the number of measurements required to achieve a certain accuracy in the relative error of the estimate of an expectation value depends on the state (Eq. 8), even when using other methods like de-randomized measurements (where $M \geq 1/(\text{Tr}[\rho O]^2 \epsilon^2) \log(2/\delta)$) [5] or Bell measurements on two copies of the state (where $M \geq 1/(\text{Tr}[\rho O]^4 \epsilon^4) \log(2/\delta)$) [6]. This is consistent with the general expectation in quantum simulation that the relative error arising from imperfect simulation is much larger than the absolute error [8].

## 3.3 Can we use shadows for time evolution?

As the classical shadow is a considerably more compact description of a given quantum state, it is interesting to ask if it can be time evolved to obtain a useful description of the quantum state at a later time. Colloquially, how well does the time evolved shadow approximate the shadow of the time evolved state?

Consider the time evolution of a state $\rho$ with a time varying unitary $V(t)$, generated from some underlying local dynamics through a Hamiltonian or a quantum circuit, which

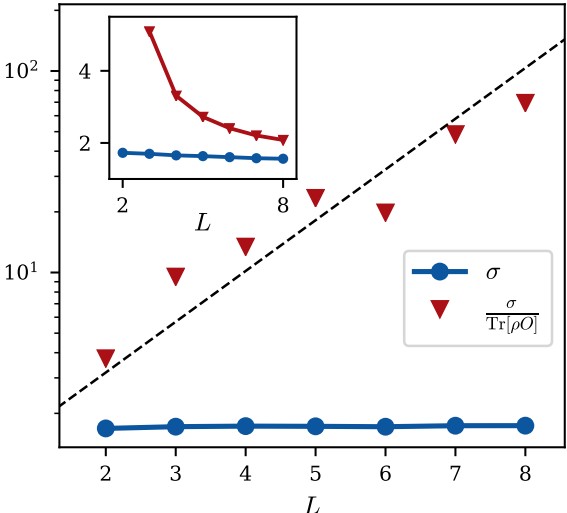

Figure 1: Relative error in the estimate of expectation value of $O = Z_i$, denoted by $\hat{o}$, for a typical Haar random state (red triangles), with the standard deviation being roughly constant (blue circles), as the system size $L$ varies. The standard deviation of $\hat{o}$ is computed using $10^4$ samples $\{\hat{o}_m\}$ generated from as many snapshots, for each $L$. Since the exact expectation value decreases exponentially in $L$, we see a roughly exponential increase in the relative error (with the dashed line as exponential guide). In the inset, the relative error is plotted for $O = Z_i$, with the state given by $|\psi\rangle = \frac{1}{\sqrt{L}}(|0111\ldots\rangle + |1011\ldots\rangle + \ldots)$, such that the exact expectation value is $\langle Z_1 \rangle = 1 - 2/L$.

leads to the time evolved expectation value $\text{Tr}[O\rho(t)] = \text{Tr}[V^\dagger(t)OV(t)\rho(0)]$ for a given operator $O$. We would like to compare this to the time dependent estimates obtained via evolving the shadow $\{\rho_s(0)\}$ of the state at $t = 0$. Each inverted snapshot $\hat{\rho}_m$ evolves as $\hat{\rho}_m(t) = V(t)\hat{\rho}_m V^\dagger(t)$, and hence our estimate of expectation value of $O$ evolves as $\frac{1}{M}\sum_m \text{Tr}[O\hat{\rho}_m(t)] = \frac{1}{M}\sum_m \text{Tr}[V^\dagger(t)OV(t)\hat{\rho}_m]$, From this we learn that the absolute error increases with time according as operator $O$ spreads in time. Thus if $O$ evolves to contain dominantly $k(t)$ body operators at time $t$, we expect that the bound of the variance computed with the time evolved shadow will grow as

$$\text{var}(\text{Tr}[O\hat{\rho}(t)]) \leq 4^{k(t)}. \tag{9}$$

Thus depending on whether the dynamics are chaotic or many body localized or non interacting, the variance in the estimate of the expectation value grows exponentially ($k(t) \sim t$) or polynomially ($k(t) \sim \log(t)$) or neither ($k(t) \sim O(1)$) [9–12].

In contrast the expectation value estimated by the classical shadow of the exact $\rho(t)$ will be bounded by a "fixed" variance $\sim 4^{k(0)}$. Of course as we noted earlier, the absolute error does have a weak state and, therefore, time dependence and the relative error will have a potentially much stronger time dependence. Nevertheless our analysis here indicates that time evolved shadows do not provide very good estimates of time dependent expectation values except in the case of non-interacting systems where they are not needed anyway. Bell measurements on two copies of the state can be used to estimate the absolute expectation value of an observable $O$ in a time evolved state, such that $|\text{Tr}[O\rho(t)]| = |\text{Tr}[V^\dagger(t)OV(t)\rho(0)]|$, whenever

$O(t) = V^\dagger(t)OV(t)$ is still a Pauli string, or equivalently, $V(t)$ is a Clifford unitary. For such an evolution, $\text{var}(|\hat{o}(t)|) \leq 1$, and the number of measurements required ($M \geq \frac{1}{\epsilon^4}\log(2/\delta)$) does not depend on $t$, unlike Eq. 9. However, for generic/non-Clifford time evolution, $O(t)$ does not remain a Pauli string and the estimation procedure of Ref. [6] ceases to be applicable.

## 3.4 Over-completeness

The representation of the density matrix in (2) has the form of an expansion of the density matrix in a basis of operators. In (2) the coefficients in the expansion are Born rule probabilities for measurement outcome $b$ after the application of unitary $U$. However we now note that *as an expansion* (2) is not unique as the operator basis is overcomplete.

Let us spell this out in the simplest case, that of a single qubit state and the Haar random unitary ensemble. In this case (2) takes the explicit $2 \times 2$ matrix form

$$\rho = \int \frac{d\mathbf{n}}{4\pi} \sum_{m=-1,1} p(m,\mathbf{n})(1 + 3m\boldsymbol{\sigma}.\mathbf{n}). \tag{10}$$

This may be re-expressed as

$$\rho = \int \frac{d\mathbf{n}}{4\pi} p_+(\mathbf{n}) + 3\sigma_j \int \frac{d\mathbf{n}}{4\pi} p_-(\mathbf{n})n_j, \tag{11}$$

where $p_\pm(\mathbf{n}) \equiv p(1,\mathbf{n}) \pm p(-1,\mathbf{n})$. For a fixed $\rho$, there are many functions $p_\pm(\mathbf{n})$ which satisfying this equation. Indeed, it is easy to show that Eq. 11 fixes only the $l = 0$ spherical harmonic of $p_+$ and the $l = 1$ spherical harmonic of $p_-$ ; all other harmonics are can be freely tuned while respecting Eq. 10. This lack of uniqueness of the expansion coefficients in Eq. 10 is not restricted to Haar random unitaries, but is applicable even when Clifford unitaries are used.

## 4 Hybrid Shadows

We now consider a theoretical generalization of the notion of the classical shadow to a hybrid classical-quantum shadow: We measure some of the qubits but not all and keep an entangled quantum state on the remaining qubits (Figure 2). The hybrid shadow can require fewer measurement rounds to achieve accurate estimates for certain observables (to be specified). On the other hand, the resulting hybrid shadow is comprised of a collection of states which will tend to have less entanglement (and require less memory to store) than the initial quantum state. Hybrid shadows may therefore provide a useful method for compressing a quantum state on a classical computer.

To form a hybrid shadow from initial state $\rho$, pick and fix a subsystem $A$ (consisting of $L_A$ qubits), denoting its complement $B$. Rotate by a random unitary on $A$ denoted by $U_A$, and then measure $A$ in the computational basis. The resulting snapshot is $U_A^\dagger |a\rangle \langle a| U_A \otimes \rho_B(a)$, where $|a\rangle$ is the state (on subsystem $A$) after measurement in the computational basis, and $\rho_B(a)$ is the resulting unnormalized collapsed state on the remainder of the qubits which are not measured (not the reduced density matrix). The distribution of such collapsed states for many body chaotic systems is studied in [13]. The probability of measuring $|a\rangle$ is given by $p = |(\langle a| \otimes \mathbf{I})|\psi\rangle|^2 = \text{Tr}[(|a\rangle \langle a| \otimes \mathbf{I})U_A \rho U_A^\dagger]$. Averaging over unitaries and measurement outcomes induces a quantum channel on the density matrix

$$\int dU \sum_{a\epsilon A} \left(U_A^\dagger |a\rangle \langle a| U_A\right) \otimes \left((\langle a| \otimes \mathbf{I})U_A \rho U_A^\dagger(|a\rangle \otimes \mathbf{I})\right) = \mathcal{M}(\rho), \tag{12}$$

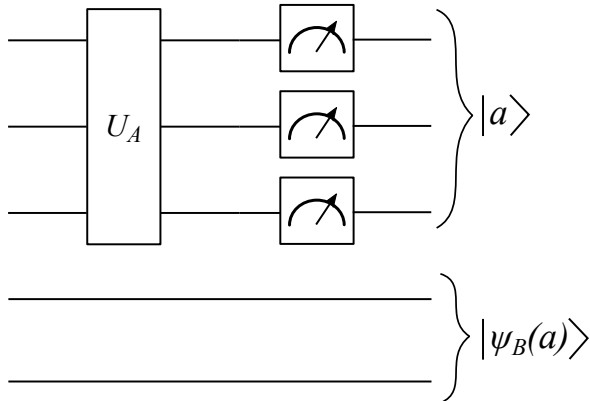

Figure 2: Measurement scheme to obtain a hybrid shadow of the state $|\psi\rangle$. A random subsystem $A$ is chosen onto which a random unitary is applied. The same subsystem is then measured in the computational basis, with the measurement outcome $|a\rangle$ and the collapsed state on the complementary subsystem $|\psi_B(a)\rangle$ (or $\rho_B(a)$ if starting with a mixed state). Upon rotating back with the unitary, a snapshot $U_A^\dagger |a\rangle \langle a| U_A \otimes \rho_B(a)$ defines a map starting from $\rho$ to itself. This map, when invertible, provides us a hybrid shadow, as described in Eq. 13, which reproduces the initial state upon averaging over the measurement outcomes, random unitaries, and the choice of subsytem $A$.

where the channel $\mathcal{M}$ is a superoperator acting only on subsystem $A$, and it may be invertible depending on the unitary ensemble. When it is invertible, a hybrid classical shadow is comprised of inverted snapshots

$$\hat{\rho} = \mathcal{M}^{-1}\left[U_A^\dagger \,|\, a\rangle\langle a \,|\, U_A\right] \otimes \rho_B(a). \tag{13}$$

For unitaries which are acting on each site and are either Clifford/Haar random unitaries, the inverse map is simply

$$\mathcal{M}^{-1}(U_A^\dagger |a\rangle \langle a| U_A) = \bigotimes_{j \in A} \left(3 U_j^\dagger |a_j\rangle \langle a_j| U_j - \mathbf{I}_{2 \times 2}\right), \tag{14}$$

implying that the corresponding part of the hybrid shadow in Eq. 13 can be efficiently stored and reconstructed on a classical computer. Similar to the classical shadow, $\mathrm{Tr}[\hat{\rho}] = 1$, however $\hat{\rho}$ is not positive semi-definite.

With this inverse map, the bound on the variance of a Pauli string observable $O$ with weight $k$ can be shown to be[2]

$$\mathrm{var}(\hat{o}) \le 3^{k(O_A)} - \rho(O)^2, \tag{15}$$

with $k(O_A)$ being the weight of the part of $O$ on subsystem $A$. When $k(O_A) = 0$, even if $k(O)$ is large, the variance is minimal and is given by $1 - \rho(O)^2$, and is demonstrated in Figure 3. On the other hand, note that the variance in Eq. 15 is at worst exponential in $L_A$; in contrast, the variance in the original shadow procedure (Eq. 6) could be exponential in total system size for long Pauli strings $O$. This means that hybrid shadows can lead to a reduction in the error in estimates for expectation values according to Eq. 5, when $L_A$ is smaller than $L$. Alternatively, if one knows the support of Pauli string $O$ beforehand, hybrid shadows can lead to significantly reduced error by choosing $A$ such that $k(O_A)$ is minimal. Note that the improvement in variance by reducing $L_A$, or increasing $L - L_A$, comes at the cost of exponentially larger classical memory required to store the collapsed states on $B$.

---

[2]See supplementary material.

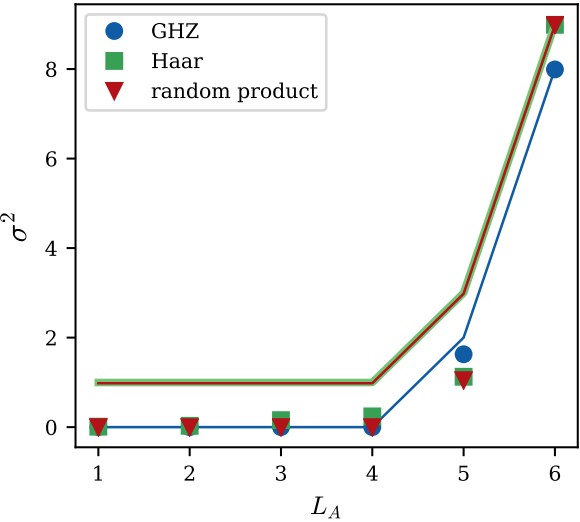

Figure 3: Variance of the estimated expectation value $\hat{o}$ of the Pauli string $O = Z_i Z_j$. To compute this, we form $10^6$ inverted hybrid snapshots Eq. (13), and use these to form the same number of estimates $\{\hat{o}_m\}$ for $O$. We plot the variance of this sequence of estimates, which should in turn estimate the left hand side of Eq. (15). We calculate the variance for a typical random product state (red triangles), a typical Haar state (green squares), and a GHZ state (blue dots) with $L = 6$, as $L_A$ is varied. Subsystem $A$ is measured and is always taken to be the first $L_A$ qubits and $i = 5, j = 6$. The solid lines of the corresponding color represent the bound on the variance in Eq. 15, and is consistent with the computed variance for these states.

## 4.1 Averaging over random choices of $A$

If information about $O$ is not available beforehand, an optimal choice of $A$ cannot be predetermined. Therefore, for such general cases $A$ can be randomly chosen as well. Picking $L_A$ qubits of $A$ at random with a probability $p(A)$, the map $\mathcal{M}(\rho)$ can be written as

$$\sum_A p(A) \int dU \sum_{b \epsilon A} U_A^\dagger |a\rangle \langle a| U_A \otimes (\langle a| \otimes \mathbf{I}) U_A \rho U_A^\dagger (|a\rangle \otimes \mathbf{I}) = \sum_A p(A) \mathcal{M}_A(\rho), \quad (16)$$

where by linearity of quantum channels, the term on the right is also a quantum channel.

The variance of the estimate for the expectation value of a Pauli string observable with weight $k$ can be obtained from Eq. 15,[3] and depends on the relation between $k$ and $L_A$. When $L_A \gg k$, $\mathrm{var}(\hat{o}) \le 3^k - \rho(O)^2$, as expected for example when all qubits are measured, on the other hand, when $k \gg L_A$, the upper bound on variance is only dependent on $L_A$, as when $A$ is fixed, given by $3^{L_A} - \rho(O)^2$. For intermediate cases the bound can also be non-exponential, for example when $L_A = 1$, $\mathrm{var}(\hat{o}) \le 1 + 2k/L - \rho(O)^2$, while when $k = 1$, $\mathrm{var}(\hat{o}) \le 1 + \frac{2L_A^2}{L + L_A(L_A - 1)} - \rho(O)^2$.[3]

---

[3]See supplementary material.

## 4.2 Time evolution and state dependence

The dependence of the variance in Eq. 15 prompts us to ask as an extension of the question about time evolution in section 3.3 - how well does the time evolved hybrid shadow approximate the initial state when compared to the classical shadow. To that end, note again that the estimate of expectation value of $O$ evolves according to how the operator spreads in time, since $\frac{1}{M} \sum_m \text{Tr}[O\hat{\rho}_m(t)] = \frac{1}{M} \sum_m \text{Tr}[V^\dagger(t)OV(t)\hat{\rho}_m]$, where $\hat{\rho}_m$ are now the inverted hybrid snapshots in Eq. 13. Thus, using Eq.15, we can say that the bound on the variance of the expectation value of $O(t)$ only depends on the weight of the observable $O(t)$ on $A$. Consider the example when initially $O$ has no weight on $A$. In this case, until the time that the front of the spreading operator reaches from its original position in $B$ to $A$, the bound on the variance remains $1 - \rho(O)^2$, and starts to increase exponentially only after that, until again when the operator has fully spread in $A$, after which the bound on the variance saturates as $\sim 4^{L_A}$. This is in contrast to time evolving classical shadows, which after saturation according to Eq. 9 will accumulate exponential error in the full system size $L$. When the choice of $A$ is random, we expect a similar saturation with $L_A$, however the initial growth in the bound would start right away, as per the discussion in the previous section.

While the bounds on absolute error are only weakly dependent on the state, actual errors in the estimates of observables can provide more insight into the nature of hybrid shadows. To that end, we show in Figure 4, the dependence of the variance of the estimate of expectation value of $X_i$ for the ground state of transverse field Ising model, with the Hamiltonian $H = -\sum Z_i Z_{i+1} - g \sum X_i$, for $L = 8$ as the transverse field $g$ is varied, for different $L_A$ ($A$ chosen randomly), for a fixed number of samples ($\approx 10^5$). The variance is lower in the paramagnetic regime for all $L_A$, and the change is most distinct when $L_A = L$, due to $\mathbb{E}_{U,|a\rangle} \hat{o}^2$ being independent of state, and the only dependence with $g$ comes from $(\mathbb{E}_{U,|a\rangle} \hat{o})^2$, where $\mathbb{E}_{U,|a\rangle}$ denotes the average over measurements, unitaries and choice of $A$ (a similar argument for $O = Z$ would mean an increase in error with $g$). As $L_A$ is decreased the variance reduces as well (see discussion below Eq. 15), which makes sense since the shadow state becomes closer to the true state as $L_A$ is reduced to 0. The dependence of the variance with with $g$ also becomes weaker. The maximum value of 3 when $L_A = L$ is also consistent with the bound in Eq.6. Together, they illustrate that the error in shadow estimates is sensitive to the phase of the ground state, and the number of qubits measured $L_A$.

## 4.3 State approximation/compression

Hybrid shadows with randomly selected $L_A$ qubits can serve as a potential state approximation tool. While a general quantum state on $L$ qubits requires a memory scaling as $2^L$, hybrid shadows could in certain circumstances compactly approximate the quantum state. Storing a collection of $M$ snapshots of the system Eq. (13) would require memory $M(L_A + d_B)$, where $d_B$ is the typical memory required to store the residual state on $B$, $\rho_B$. In the worst case scenario where the residual state is highly entangled, $d_B = 2^{L_B}$. On the other hand if the residual states have area law entanglement, then $d_B = \mathcal{O}(L_B)$ (scaling of number of parameters required to store an associated matrix product state in 1D).

The hybrid shadow becomes a better approximation to the original state as the number of samples in the shadow $M$ increases. As before, the convergence with $M$ is controlled by the snapshot-to-snapshot variance in the shadow estimate of the observable in question Eq. (15). It follows that the error in a Pauli observable $O$ can be made $\epsilon$ small with probability $\geq 1 - \delta$ by insisting that $M \geq C \frac{3^{k(O_A)}}{\epsilon^2} \log(2/\delta)$.

The ratio of total classical memory required to store the shadow states and the classical memory required to store the entire state is approximately given by $M(L_A + d_B)/2^L$, and when the expectation values of Pauli strings of weight $k(O_A)$ on A are desired up to an accuracy $\epsilon$

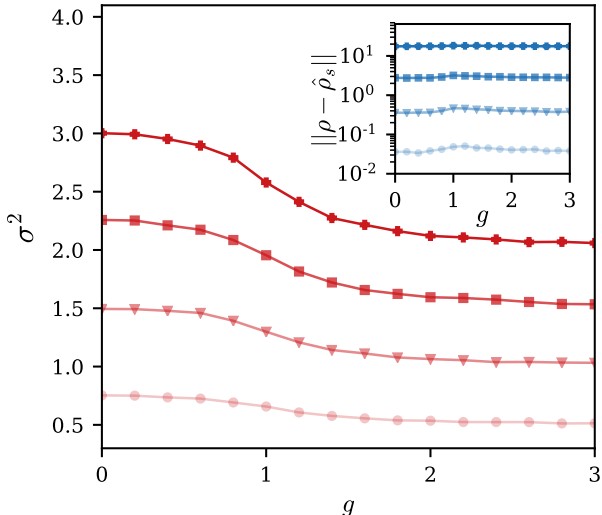

Figure 4: Variance of the estimate of expectation value $\hat{o}$ for $O = X$ on a single qubit. This is calculated using $10^5$ inverted hybrid snapshots (Eq. 15) to generate as many estimates $\{\hat{o}_m\}$ of the observable $O$. Each snapshot is formed using a random choice of subset $A$ of fixed size $L_A$. The procedure is carried out on the ground state of the transverse field Ising model ($H = -\sum Z_i Z_{i+1} - g \sum X_i$, $L = 8$) vs the transverse field $g$, for $L_A = 2, 4, 6, 8$ (increasing in darkness), implying the error in estimates are sensitive to $L_A$ as well as the phase of the ground state. Inset: The $L_1$ norm of the the difference between the estimated state $\hat{\rho}_s = \frac{1}{M} \sum_m \hat{\rho}_m$, using $M = 10^5$ inverted hybrid snapshots $\hat{\rho}_m$ for each point, and the true state $\rho$, vs $g$ for $L_A = 2, 4, 6, 8$ shows little dependence on $g$, and increases exponentially with $L_A$.

with probability $1 - \delta$, the ratio scales as $3^{k(O_A)} \frac{(L_A + d_B)}{2^L} \log(2/\delta)/\epsilon^2$ in the large $L$ limit. For small $L_A$, $d_B \sim 2^L$, and low errors in the estimate of expectation values, the ratio can be quite large in practice. When $k$ is small compared to $L_A$ the hybrid shadow approximation appears to be more memory efficient than storing the full state. On the other hand, when compared to the fully classical shadow ($L_A = L, d_B = 0$) more classical memory is required for storing a state using hybrid shadows, since the ratio of the memory required to classically store a hybrid shadow and a classical shadow is given by $3^{-k(O_B)} \frac{(L_A + d_B)}{L}$. For arbitrary $O$, the ratio is $(L_A + d_B)/L$ and will generally be larger than one, but can be improved either by using approximation techniques for low entanglement states (such as a matrix product states) or by using standard compression techniques for arbitrary states [14, 15]. Note that when a limited capacity quantum memory is available, the collapsed state on $B$ can be stored optimally in a hybrid classical quantum memory, which can be another potential application of hybrid shadows.

Indeed, it is interesting to ask if there are classes of quantum states for which this thinning out of the quantum degrees of freedom stored in full, preserves greater accuracy than for other classes for which it does not. As a first test we have examined the ground states of the transverse field Ising model as the quantum fluctuations are varied. As is well known, the entanglement in the ground states peaks at the critical point. Somewhat surprisingly, in contrast to traditional state approximation methods like the matrix product states, there is remarkably little dependence on the entanglement of the state. This is supported in the inset of Fig.4, by computing the $L_1$ norm $||\rho - \hat{\rho}_s||$ for the estimated state $\hat{\rho}_s$ using hybrid shadows

for the transverse field Ising model, keeping the number of inverted snapshots $M = 10^5$ for all $g$. The norm appears largely independent of g, especially for larger $L_A$, whereas for a matrix product state it is larger near the critical regime for fixed bond dimensions. However, analogous to Eq. 15 when generalized to global operators, reducing $L_A$ leads to an exponential reduction in the norm for a fixed number of snapshots.

## 5 Discussion

We have commented on certain aspects of classical shadows from the viewpoint of many body physics. Unlike absolute errors in estimates of expectation values, relative errors can have a strong state dependence. Further, time evolution performed using classical shadows leads to an exponential buildup of errors in a generic many body system. Along the way, we also pointed out the freedom in writing down the expansion of a state using classical shadows. Finally, we introduced the idea of forming a hybrid quantum-classical shadow from measurements performed over some of the qubits instead of all, which can more accurately predict expectation values at the expense of requiring more quantum memory. These hybrid shadows could provide a way of thinking about the manner in which correlations of the state are captured, as more and more number of experiments are used to compute the shadow, and how it contrasts with traditional ways in which states are approximated. Such states could also potentially enhance the capabilities of near term quantum computers with limited number of qubits.

## Acknowledgements

The authors would like to thank Robert Huang and Max McGinley for useful discussions. CvK is supported by a UKRI Future Leaders Fellowship MR/T040947/1. This work was supported by a Leverhulme Trust International Professorship grant number LIP-202-014 (SLS). For the purpose of Open Access, the author has applied a CC BY public copyright licence to any Author Accepted Manuscript version arising from this submission.

## A Hybrid shadows

Here we derive the results for hybrid shadows in greater detail. Consider starting with a state $\rho$ defined on $L$ qubits, and partitioning the system into subsystems $A$ and $B$ with number of qubits $L_A$ and $L_B$. $\rho$ is rotated using a random unitary $U_A$ acting on $A$ such that the new state is $U_A \otimes \mathbf{I_B} \, \rho \, U_A^\dagger \otimes \mathbf{I_B}$, where $\mathbf{I_B}$ is the identity on $B$. A measurement is then performed on $A$ in the computational basis, resulting in the collapsed state $|a\rangle$. The *un-normalized* collapsed state on $B$ is then given by

$$\rho_B(a) = (\langle a| \otimes \mathbf{I_B})U_A \otimes \mathbf{I_B} \, \rho \, U_A^\dagger \otimes \mathbf{I_B}(|a\rangle \otimes \mathbf{I_B}), \tag{A.1}$$

$$\rho_B(a) = (\langle a| U_A \otimes \mathbf{I_B}) \, \rho \, (U_A^\dagger |a\rangle \otimes \mathbf{I_B}). \tag{A.2}$$

The normalized state on the full system can then be written as $|a\rangle \langle a| \otimes \rho_B(a)/\text{Tr}_B[\rho_B(a)]$. A snapshot of $\rho$ is the state obtained after rotating back the subsystem $A$: $U_A^\dagger |a\rangle \langle a| U_A \otimes \rho_B(a)$. Note that the collapsed state on $B$ is unchanged in this step. When averaged over the unitary

ensemble and the measurement outcomes $|a\rangle$, $\mathcal{M}(\rho)$ defines a map acting on $\rho$ via

$$\int dU \sum_{a \epsilon A} U_A^\dagger |a\rangle \langle a| U_A \otimes \frac{\rho_B(a)}{\text{Tr}_\text{B}[\rho_B(a)]} \text{Tr}_\text{B}[\rho_B(a)] = \mathcal{M}(\rho), \tag{A.3}$$

such that the probability of measuring $|a\rangle$ cancels the normalization and the map only depends explicitly on the un-normalized state $\rho_B(a)$. In general, finding an inverse of the map $\mathcal{M}$ to reconstruct the state $\rho$ is a difficult task, however, whenever the the unitaries are constructed that the classical shadow can be formed (or when all the qubits are measured), we can show that a hybrid shadow can be written down as

$$\hat{\rho}_m = \mathcal{M}_A^{-1}(U_A^\dagger |a\rangle \langle a| U_A) \otimes \frac{\rho_B(a)}{\text{Tr}_\text{B}[\rho_B(a)]}, \tag{A.4}$$

noting that no operation is performed on the collapsed state on $B$. To see that $\hat{\rho}_m$ reproduces $\rho$ in expectation, expand $\rho = \sum_r c_r \rho_A^r \otimes \rho_B^r$, so that upon averaging the classical shadow on $A$ reproduces $\rho_A^r$,

$$\left( \int dU_A \sum_{a \epsilon A} \mathcal{M}_A^{-1}(U_A^\dagger |a\rangle \langle a| U_A) \ \langle a| U_A \rho_A^r U_A^\dagger |a\rangle \right) \otimes \rho_B^r = \rho_A^r \otimes \rho_B^r, \tag{A.5}$$

and by linearity $\rho$ is reproduced. For the case when $U_A$ is a product of on site Haar random unitaries $U_i$, which we have specialized to in the main text, the hybrid shadow has the form

$$\hat{\rho}_m = \left[ \otimes_{i \epsilon A} \left( 3 U_i^\dagger |a_i\rangle \langle a_i| U_i - \mathbf{I}_i \right) \right] \otimes \frac{\rho_B(a)}{\text{Tr}_\text{B}[\rho_B(a)]}. \tag{A.6}$$

## A.1 Variance of expectation values

First, let's consider the case when a fixed subsystem $A$ is measured, and the Pauli string $O$ has a decomposition $O = O_A \otimes O_B$, and the expectation $\rho(O)$ in state $\rho$. Denoting the estimate for the expectation value as $\hat{o} \equiv \text{Tr}(\hat{\rho}O)$, and the exact value as $\rho(O)$, the variance can be written down as

$$\text{var}(\hat{o}_m) = \left( \int dU_A \sum_{|a\rangle} \left( \text{Tr}\left[ \mathcal{M}_A^{-1}(U_A^\dagger |a\rangle \langle a| U_A) O_A \right] \otimes \text{Tr}\left[ \frac{\rho_B(a)}{\text{Tr}_\text{B}[\rho_B(a)]} O_B \right] \right)^2 \text{Tr}_\text{B}[\rho_B(a)] \right) - \rho(O)^2, \tag{A.7}$$

$$\text{var}(\hat{o}_m) = \left( \int dU_A \sum_{|a\rangle} \left( \text{Tr}\left[ \mathcal{M}_A^{-1}(U_A^\dagger |a\rangle \langle a| U_A) O_A \right] \right)^2 \text{Tr}_\text{B}[\rho_B(a)] \right) \left( \text{Tr}\left[ \frac{\rho_B(a)}{\text{Tr}_\text{B}[\rho_B(a)]} O_B \right] \right)^2 - \rho(O)^2, \tag{A.8}$$

$$\text{var}(\hat{o}_m) \leq \int dU_A \sum_{|a\rangle} \left( \text{Tr}\left[ \mathcal{M}_A^{-1}(U_A^\dagger |a\rangle \langle a| U_A) O_A \right] \right)^2 \text{Tr}_\text{B}[\rho_B(a)] - \rho(O)^2, \tag{A.9}$$

$$\text{var}(\hat{o}_m) \leq 3^{k(O_A)} - \rho(O)^2, \tag{A.10}$$

where in Eq. A.9 we have used the assumption of Pauli string $O$ so that $\rho(O) \leq 1$, whereas the last line can be derived either by assuming Haar random unitaries acting on each qubit and the explicit form for $\mathcal{M}^{-1}$, or for general unitaries using the "shadow norm" defined in [2].

When the $L_A$ qubits are randomly chosen, the estimate of the state $\rho$ can be written in terms of $\binom{L}{L_A}$ independent random variables $\hat{\rho}_s = \frac{1}{M} \sum_{M,A} p(A) \hat{\rho}_m(A)$, such that $p(A)$ is the probability of choosing subsystem $A$, which by linearity still averages out to the exact state. The variance can then be estimated by the weighted sum of variances

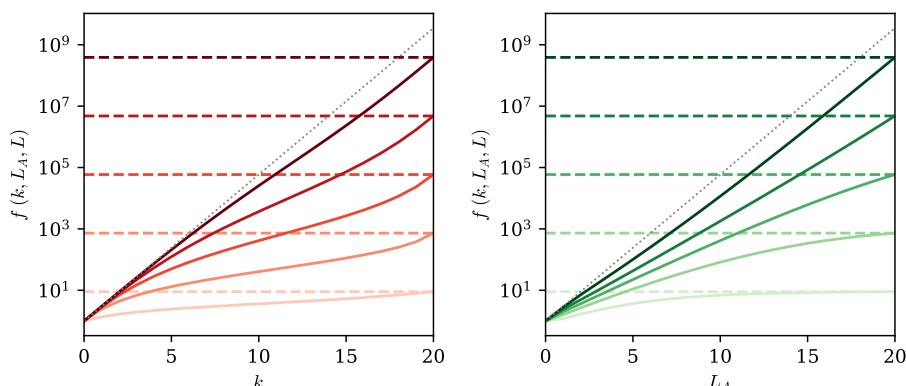

Figure 5: The function $f(k, L_A, L)$ in Eq. A.12 determining the upper bound of the variance of the expectation value of a weight $k$ Pauli string using hybrid shadows with randomly selected subsystem $A$, for $L = 20$. Left - $f(k, L_A, L)$ vs $k$ for $L_A = 2, 6, 10, 14, 18$ (increasing in darkness), with the horizontal dashed lines showing saturation at $3^{L_A}$. The bound for the full classical shadow (Eq. 5) is plotted as the dotted line. Right: $f(k, L_A, L)$ plotted vs $L_A$ for $k = 2, 6, 10, 14, 18$ (increasing in darkness). The horizontal dashed lines are the bounds for the classical shadow, when $L_A = L$. As $k$ increases the curves approach the dotted line is $3^{L_A}$.

$$\text{var}(\hat{o}_m) \leq \sum_A p(A) 3^{k(O_A)} - \rho(O)^2, \tag{A.11}$$

where

$$\sum_A p(A) 3^{k(O_A)} = f(k, L_A, L)$$

$$= \left( \sum_{\max(0, k-L+L_A)}^{\min(k, L_A)} \binom{k}{r} \binom{L_A}{r} \binom{L-k}{L_A - r} 3^r \right) / \left( \sum_{\max(0, k-L+L_A)}^{\min(k, L_A)} \binom{k}{r} \binom{L_A}{r} \binom{L-k}{L_A - r} \right), \tag{A.12}$$

is obtained by solving the combinatorially equivalent problem of distributing $L_A$ distinct objects into 2 boxes of size $k$ and $L - k$ without replacement. The function $f(k, L_A, L)$ which is the first term on the right in the second equation is plotted for $L = 20$ and different $k$ and $L_A$. For $L_A = 1$, it goes as $1 + 2k/L - \rho(O)^2$, for $k = 1$ as $1 + \frac{2L_A^2}{L + L_A(L_A - 1)} - \rho(O)^2$, whereas for $L_A \gg k$ the upper bound scales as $3^k - \rho(O)^2$, and when $k \gg L_A$ it scales as $3^{L_A} - \rho(O)^2$. Excluding the exact value on the right, the bound on variance is plotted for different $k, L_A$ for $L = 20$ in Figure 5.

# B  Classical and hybrid shadows from global unitaries

Here we provide generalizations of our results when global Clifford or Haar random unitaries $U$ are applied on an unknown state $\rho$ before performing measurements, followed by forming inverted snapshots. As noted in the main text, the inverted snapshot when all of the qubits are measured is given by

$$\mathcal{M}^{-1}(U^\dagger |b\rangle \langle b| U) = (2^L + 1) U^\dagger |b\rangle \langle b| U - \mathbf{I}_{2^L \times 2^L}. \tag{B.1}$$

It was shown in [2] that using these inverted snapshots to estimate the expectation value of a traceless operator $O$ leads to a variance in the estimate $\hat{o}_m$ given by

$$\text{var}(\hat{o}_m) = \left( \int dU \sum_{|b\rangle} \left( \text{Tr}\left[ \mathcal{M}^{-1}(U^\dagger |b\rangle \langle b| U)O \right] \right)^2 \right) - \rho(O)^2 \,, \tag{B.2}$$

$$\text{var}(\hat{o}_m) = \frac{2^L + 1}{2^L + 2} \left( \text{Tr}\left[ O^2 \right] + 2\text{Tr}\left[ \rho O^2 \right] \right) - \rho(O)^2 \,, \tag{B.3}$$

$$\text{var}(\hat{o}_m) \le 3\text{Tr}[O^2] - \rho(O)^2 \,, \tag{B.4}$$

where in Eq. B.4 $\text{Tr}[O^2]$ is the squared Hilbert-Schmidt norm of the operator $O$. Thus, whenever $\text{Tr}[O^2]$ is bounded and small, quantities such as the fidelity of the state (for which $O = \rho - \frac{1}{2^L}\mathbf{I}$, $\text{var}(\hat{o}_m) = O(1)$) can be efficiently estimated using $M \ge C\frac{\text{var}(\hat{o})}{\epsilon^2}\log(2/\delta)$ samples with an accuracy $\epsilon$ and probability $1 - \delta$, independent of $L$.

However, when estimating the expectation value of a Pauli string $O$, the variance scales with the system size $L$, independent of the weight of the Pauli string (ie. the number of qubits where the operator is not identity), such that $\text{var}(\hat{o}_m) = 2^L + 1 - \rho(O)^2$, obtained from Eq. B.3. This makes using this approach highly inefficient when compared to using local unitaries to form the shadow, where the variance only depends on the weight $k$ of the Pauli string (Eq. 6). Based on this observation, we can arrive at the following conclusion about state dependence and time evolution:

**State dependence of absolute and relative error:** Limiting ourselves to the case of $O$ being a Pauli string, the relative error for the estimate of the expectation value of $O$ has a dependence on the state given by

$$\frac{\text{var}(\hat{o})}{\text{Tr}[\rho O]^2} = \left( \frac{2^L + 1}{\text{Tr}[\rho O]^2} - 1 \right) \,, \tag{B.5}$$

where we have used Eq. B.3. By virtue of the exponential scaling in the numerator, it is not as sensitive to whether $\text{Tr}[\rho O] = e^{-\mathcal{O}(L)}$ for a Haar random state or $\text{Tr}[\rho O] = O(1)$ for a product state, when compared to local shadows, but the relative error itself is exponentially larger (Eq. 7).

**Time evolution:** Do shadows produced from global unitaries offer any advantage over shadows obtained from local unitaries, when considering estimating the expectation value of a Pauli string $O$, in a time evolved state generated from inverted snapshots $\hat{\rho}_m(t) = V(t)\hat{\rho}_m V^\dagger(t)$ using the time evolution operator $V(t)$? As discussed in Sec. 3.3, the estimate of the expectation value of $O$ evolves as $\frac{1}{M}\sum_m \text{Tr}[O\hat{\rho}_m(t)] = \frac{1}{M}\sum_m \text{Tr}[V^\dagger(t)OV(t)\hat{\rho}_m]$, and for local shadows the absolute error increases with time according to how operator $O$ spreads in time (Eq. 9). On the other hand, for global shadows, the variance does not depend on the weight of the Pauli string $O$ (Eq. B.4), and is given as

$$\text{var}(\hat{o(t)}_m) \sim \text{var}(\hat{o}_m) \sim \exp(L)\,. \tag{B.6}$$

Thus, even though the error does not change much with time, it scales exponentially in the system size $L$ at all times.

**Hybrid shadows:** Hybrid shadows obtained by applying a unitary $U_A$ on subsystem $A$ comprising of $L_A$ qubits out of a total $L$ qubits can be readily generalized from Sec 4 when $U_A$

acting on $A$ is a Clifford or Haar random unitary. When the measurement outcome on $A$ is $|a\rangle$, the hybrid shadow in Eq. A.4 can be simplified to

$$\hat{\rho}_m = \left[ (2^{L_A} + 1) U_A^\dagger |a\rangle \langle a| U_A - \mathbf{I}_{2^{L_A} \times 2^{L_A}} \right] \otimes \frac{\rho_B(a)}{\mathrm{Tr}_B[\rho_B(a)]} \,. \tag{B.7}$$

The variance of the estimate of expectation value of a Pauli string $O = O_A \otimes O_B$ can then be derived as follows

$$\mathrm{var}(\hat{o}_m) = \left( \int dU_A \sum_{|a\rangle} \left( \mathrm{Tr}\left[ \mathcal{M}_A^{-1}(U_A^\dagger |a\rangle \langle a| U_A) O_A \right] \otimes \mathrm{Tr}\left[ \frac{\rho_B(a)}{\mathrm{Tr}_B[\rho_B(a)]} O_B \right] \right)^2 \mathrm{Tr}_B[\rho_B(a)] \right) - \rho(O)^2 \,, \tag{B.8}$$

$$\mathrm{var}(\hat{o}_m) \leq \int dU_A \sum_{|a\rangle} \left( \mathrm{Tr}\left[ \mathcal{M}_A^{-1}(U_A^\dagger |a\rangle \langle a| U_A) O_A \right] \right)^2 \mathrm{Tr}_B[\rho_B(a)] - \rho(O)^2 \,, \tag{B.9}$$

$$\mathrm{var}(\hat{o}_m) \leq \frac{2^{L_A} + 1}{2^{L_A} + 2} \left( \mathrm{Tr}[O_A^2] + 2 \right) \mathrm{Tr}_B[\rho_B(a)] - \rho(O)^2 \,, \tag{B.10}$$

$$\mathrm{var}(\hat{o}_m) \leq 2^{L_A} + 1 - \rho(O)^2 \,, \tag{B.11}$$

where in going from Eq. B.9 to Eq. B.10, we have used Eq. B.3. Comparing this to the variance when local hybrid shadows are used, in Eq. 15 ($\mathrm{var}(\hat{o}) \leq 3^{k(O_A)} - \rho(O)^2$), there is no dependence on the weight of $O_A$. This also implies that taking an average over randomly chosen subsystems $A$ doesn't reduce the upper bound of the variance like in Eq. A.12.

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
