# Peer review of "On Classical and Hybrid Shadows of Quantum States"

_SciPost Physics, doi:SciPost Phys. 14, 094 (2023)_

## Round 1 · Referee Report · Anonymous (Referee 1) · 2022-12-22

Report

I thank the authors for their revisions to the manuscript. I feel that with the changes, the paper does a very good job of putting the results in the context of other work on shadow tomography. It should be much clearer to a reader with expertise in quantum tomography how classical shadows and other tomographic techniques might have new applications to problems in many-body physics. In addition, the work is potentially helpful for the quantum tomography community as it helps explain what types of quantities are typically of interest in many-body physics.

---

## Round 1 · Referee Report · Anonymous (Referee 2) · 2022-12-27

Strengths

  • The idea of hybrid shadows (in which one measures part of the state and uses the ensemble of quantum states corresponding to various measurement outcomes) is a new and interesting contribution to the literature.
  • The discussion of compressing quantum states and predicting time evolution through shadows is insightful.

Report

I concur with the detailed report of the previous referee. This work introduces and analyzes an interesting new concept in the area of shadow tomography. Hybrid shadows are a useful addition to the conceptual toolbox for thinking about ways of efficiently representing the information in quantum states. Hybrid shadows also potentially connect to a number of other active research topics, such as projective ensembles and measurement-induced entanglement, so this paper could potentially be influential.

---

## Round 1 · Author Response

Dear Editor and Referee,

We thank you for taking the time and reviewing our manuscript. We have prepared a revised version which hopefully addresses the Referee’s comments, and provide a detailed response to the comments below.

1) ”...The present paper does not discuss or analyze this global shadows approach and instead focuses only on local shadows without justification. This oversight leads to some erroneous or misleading conclusions in the paper as global shadows would allow the sample efficient representation of time-evolved operators. The Hilbert-Schmidt norm is invariant under unitary evolution.”

Requested change - ”provide a systematic study and analysis of global shadows. ”

We have now added an analysis of global shadows in the context of our previous results, commenting on state dependence of absolute and relative errors, time evolution and hybrid shadows, obtained in an analogous manner. Due to better implementability of local shadows on NISQ devices, we have kept the detailed discussion in an appendix, while highlighting the reason for doing so in the main text.

As for the results themselves, while we agree that global shadows can be used to efficiently estimate quantities like the fidelity, when estimating the expectation value of Pauli strings (our main focus), the variance of the estimate grows exponentially with the number of qubits, irrespective of the weight of the Pauli string. So, despite the Hilbert-Schmidt norm not varying with time under unitary evolution, for the operators typically considered in our paper, it grows with the Hilbert space dimension at all times, making it exponentially worse than local shadows.

2) ”...This limitation is well known in quantum tomography and can be overcome by instead performing Bell basis measurements on multiple-copies of the state rho. The extension of the shadow formalism to this case was provided in Huang, Kueng and Preskill PRL (2021), where it was shown how to obtain quantum advantage over single-copy classical shadows using two-copy measurements. It is a very simple, but powerful idea. The operators P x P are commuting for any Pauli P, therefore one can measure |Tr[P ρ]|^2 efficiently using Bell basis measurements. ”

Requested change - ”provide appropriate commentary on the role of entangling measurements such as simple Bell basis measurements in overcoming the limitations of classical shadows. Perhaps several other results in the paper also need to be revisited in light of this perspective. ”

We have added discussions about Bell measurements in sections III.A,III.B and III.C. They have a similar state dependence of relative errors, and while they can be used to efficiently estimate expectation values under a Clifford time evolution, their utility is restricted to only measuring expectation values of Pauli strings, unlike other operations possible with classical shadows. Moreover, as mentioned in Huang, Kueng and Preskill PRL (2021), when the sign of the expectation value is desired, further measurements that require prior knowledge of O might be needed. Another disadvantage is the scaling of the number of measurements (to estimate |Tr[Oρ]|) with the accuracy ε, which is 1/ε^4 using this approach, compared to 1/ε^2 scaling for classical shadows.

Keeping these points in mind, while we do agree this method has its advantages, our paper’s aim is to focus more on aspects of classical shadow itself, so we have avoided going into too much detail of the Bell measurements.

3) ”...The hybrid shadows discussed in the present paper do not seem to have any asymptotic advantages over just using classical shadows or other standard compression techniques. Perhaps hybrid shadows could have some advantage in non-asymptotic settings, but the authors do not provide any strong evidence that this is the case. ”

Requested change - ”provide convincing evidence that hybrid shadows present some asymptotic or non-asymptotic advantage in classical simulation of quantum systems. Compare to methods based on representations of quantum states by low rank stabilizer approximations. ”

We absolutely agree that our earlier discussion on hybrid shadows did not provide any evidence for an advantage in compression when compared to classical shadows, because on its own, without compression of the collapsed quantum state, hybrid shadows will typically require more memory than classical shadows. We have emphasized this point by now considering the ratio of estimated memory for hybrid shadows and classical shadows, following the discussion based on the ratio for hybrid shadows and a complete classical description of the quantum state.

In other words, hybrid shadows need to be combined with other standard compression techniques on the collapsed state on subsystem B which is not measured, like the stabilizer approach pointed out, such that a combination of that method and the classical part provides a new approach to compression. With this paper serving as the introduction of the idea of hybrid shadows, we leave it to future works to explore the applications of hybrid shadows in more detail (Sec IV.C).

We hope our changes have improved our manuscript and thank you for your time and consideration.

Sincerely, Saumya Shivam, C. W. von Keyserlingk and S. L. Sondhi

---

## Round 1 · List of Changes

1. Added introductory paragraph on global shadows on page 2, containing new Eq. (3), and an appendix (B) titled ”Classical and hybrid shadows from global unitaries”.
  2. Discussion on Bell measurements added on page 3 (last paragraph of III.A), in context of non-random measurements, citing corresponding papers.
  3. State dependence of relative error for Bell measurements added in Sec III.B (”To summarize, the number of measurements ... (where M ≥ 1/(Tr[ρO]^4ε^4) log(2/δ)) ”.
  4. Time evolution of the estimate of expectation values using Bell measurements discussed in the end of Sec III.C (”Bell measurements...”).
  5. Comments on state compression using hybrid shadows added in the second last paragraph in Sec IV.C (”On the other hand, when compared to the fully classical shadow...”), citing additional relevant references.

---

## Editorial Decision

published